# Pseudorabies Virus Tegument Protein UL13 Suppresses RLR-Mediated Antiviral Innate Immunity through Regulating Receptor Transcription

**DOI:** 10.3390/v14071465

**Published:** 2022-07-02

**Authors:** Ningning Zhao, Fan Wang, Zhengjie Kong, Yingli Shang

**Affiliations:** 1Department of Preventive Veterinary Medicine, College of Veterinary Medicine, Shandong Agricultural University, Taian 271018, China; ningningzhao@163.com (N.Z.); wf201709@163.com (F.W.); kongzj2016@163.com (Z.K.); 2Shandong Provincial Key Laboratory of Animal Biotechnology and Disease Control and Prevention, Shandong Agricultural University, Taian 271018, China; 3Institute of Immunology, Shandong Agricultural University, Taian 271018, China

**Keywords:** pseudorabies virus (PRV), tegument protein UL13, RIG-I, MDA5, NF-κB

## Abstract

Pseudorabies virus (PRV) has evolved various strategies to escape host antiviral immune responses. However, it remains unclear whether and how PRV-encoded proteins modulate the RIG-I-like receptor (RLR)-mediated signals for immune evasion. Here, we show that the PRV tegument protein UL13 functions as an antagonist of RLR-mediated antiviral responses via suppression of the transcription of RIG-I and MDA5, but not LGP2. UL13 overexpression significantly inhibits both the mRNA and protein levels of RIG-I and MDA5, along with RIG-I- or MDA5-mediated antiviral immune responses, whereas overexpression of RIG-I or MDA5 counteracts such UL13-induced suppression. Mechanistically, UL13 suppresses the expression of RIG-I and MDA5 by inhibiting activation of the transcription factor NF-κB. Consequently, overexpression of p65 promotes the activation of *RIG-I* and *MDA5* promoters. Moreover, deletion of the p65-binding sites in the promoters of *RIG-I* or *MDA5* abolishes the suppression role of UL13. As a result, mutant PRV lacking UL13 elicits stronger host antiviral immune responses than PRV-WT. Hence, our results provide a novel functional role of UL13-induced suppression of host antiviral immunity through modulating receptors’ transcription.

## 1. Introduction

Pseudorabies virus (PRV), a member of the Alphaherpesvirus subfamily, is the pathogen of Aujeszky’s disease, which causes abortions and stillbirths in sows, central nervous system disorders in young piglets, and respiratory disease in older pigs [1,2,3], generating considerable economic losses worldwide in the swine industry worldwide [4]. Recent studies report that PRV can also infect humans, thus raising great concern about cross-species PRV transmission [5,6]. Similar to other alphaherpesviruses, PRV has evolved multiple strategies to dismantle the host’s innate antiviral response [7,8,9,10], such as blocking pattern-recognition receptor (PRR)-induced type-I interferon (IFN-I) and neutralizing the antiviral functions of IFN-stimulated genes (ISGs), resulting in the establishment of lifelong latent infection in the host. Nevertheless, viruses may also be released during latent infection, which causes persistent infection in the host, and poses potential risks to the breeding industry and human health. In addition, the constant mutation of PRV leads to classical attenuated vaccines failing to provide sufficient protection against PRV infection. Therefore, it is desirable to explore the strategies of innate immune escape of PRV for drug target selection.

As the first line of defense against viral invasion, the antiviral innate immune response is activated through the recognition of pathogen-associated molecular patterns (PAMPs) by PRRs, leading to the production of interferons (IFNs), inflammatory cytokines, and chemokines to eliminate pathogens [11,12,13]. Transmembrane or cytosol PRRs—such as Toll-like receptors (TLRs), RIG-I-like receptors (RLRs), and cGAS-STING—recognize distinct pathogen-derived nucleic acids with different features. Among them, RLRs, including retinoic-acid-inducible gene-I (RIG-I), melanoma-differentiation-associated gene 5 (MDA5), and laboratory of genetics and physiology 2 (LGP2), are prominent intracellular PRRs that normally sense RNA virus signals [12,14]. Interestingly, studies have discovered that DNA virus replication generates RNA intermediates, which could be nucleic acid ligands for RLRs [15,16,17,18]. In addition, host-derived RNAs produced by DNA viruses can also initiate the RLR-mediated signaling pathway [19]. These studies indicate that RLR-mediated antiviral innate immune response may also play important roles in defense against DNA virus infection. In response, viruses have evolved numerous strategies to evade it [16]. However, little is known about how PRV escapes from RLR-mediated signaling pathways.

PRV has a large linear double-stranded DNA genome that encodes over 70 functional proteins [20]. Among them, the tegument proteins possess a wide variety of functions in viral entry, secondary envelopment, and viral capsid transportation during infection and immune escape [21]. It has been shown that tegument proteins are the main components of alphaherpesviruses antagonizing RLR signal transduction. For instance, HSV-1 pUS11 interacts with endogenous RIG-I and MDA5 through the RNA-binding domain to block IFN-β production [22]. It has been shown that HSV-1 pUL37—a deaminase protein—suppresses RNA-induced activation by targeting RIG-I [23]. UL13, a serine/threonine protein kinase, is an important immune escape protein of PRV. Recent reports have found that the PRV tegument protein UL13 acts as an antagonist of cGAS-STING-mediated IFN-β production [24,25]. However, whether UL13 regulates the RLR pathway remains unknown. Here, we show that PRV UL13 inhibits transcription of RIG-I and MDA5 through regulating NF-κB activation, resulting in suppression of RLR-mediated IFN-β production. Our study preliminarily explores the suppressive role of the PRV UL13 in the activation of RLR-mediated antiviral innate immune response, and identifies a novel strategy of PRV for immune escape.

## 2. Materials and Methods

### 2.1. Cells and Viruses

HEK293T cells, PK-15 cells, and BHK-21 cells were purchased from the American Type Culture Collection (ATCC, Manassas, VA, USA). Stable PK-15 cells ectopically expressing PRV UL13 (UL13-PK-15 cells), along with control cells, were generated previously [24]. Cells were maintained in Dulbecco’s minimal essential medium (DMEM) (Gibco, New York, NY, USA) containing 10% fetal bovine serum (FBS) (Biological Industries, Israel) and 1% penicillin–streptomycin (Gibco, New York, NY, USA) at 37 °C in 5% CO_2_. All cells tested negative for mycoplasma using a mycoplasma detection kit (TransGen Biotech, Beijing, China). PRV (Bartha-K61 strain) was purchased from the China Veterinary Culture Collection Center (Cat# CVCC AV249, Beijing, China), and was purified in BHK-21 cells. The PRV-∆UL13 recombinant strain was generated previously [24]. The wild-type PRV (PRV-WT) and PRV-∆UL13 PRV strains were amplified and titrated in PK-15 cells using standard protocols. Sendai virus (SeV), described previously [26], was propagated in 10-day-old embryonated eggs. SeV titers were then determined via the Reed–Muench method using MDCK cells.

### 2.2. Antibodies and Reagents

The antibodies used and the sources were as follows: Anti-Flag M2 mouse mAb (1:5000, F1804) was obtained from Sigma (St. Louis, USA). Antibodies against TBK1 (1:1000, 3013), p-TBK1 (Ser172) (1:1000, 5483), p-p65 (Ser536) (1:1000, 3033), RIG-I (1:1000, 3743), MDA5 (1:1000, 5321), and MAVS (1:1000, 3993) were purchased from Cell Signaling Technology. Antibodies against p65 (1:2000, 10745-1-AP), β-actin (1:10,000, 66009-1-Ig), IκBα (1:2000, 10268-1-AP), and Lamin B1 (1:2000, 12987-1-AP) were purchased from Proteintech Group Inc. Lipofectamine 2000 transfection reagent and Opti-MEM were purchased from Thermo Fisher Scientific. Protease inhibitors were purchased from Roche. PMSF and DAPI were purchased from Solarbio Life Sciences, Beijing, China.

### 2.3. Lentiviral Infection and Stable Cell Line Generation

Stable cell lines were generated as described previously [24]. Briefly, lentiviral particles were produced in HEK293T cells transfected with two packaging plasmids (psPAX2 and pVSVG) and an empty vector or pCDH-Flag-UL13 plasmid using Lipofectamine 2000 (Invitrogen). After 24 h, the recombinant viruses were filtered, then infected HEK293T cells again and supplemented with Polybrene (6 μg/mL, Cat#H8641, Solarbio, Beijing, China). Cells were selected with puromycin (Cat# IP1160, Solarbio, China) at a final concentration of 6 μg/mL for 5 days. Monoclonal cells were obtained in 96-well plates via the limited dilution method, and UL13-HEK293T cells were identified by immunoblotting with anti-Flag monoclonal antibody (Sigma).

### 2.4. Reverse Transcription and Quantitative Real-Time PCR (qPCR)

Total RNA isolation was carried out using an RNA purification Kit (Promega, Madison, WI, USA) and reverse-transcribed to cDNA using M-MLV reverse transcriptase with RNase inhibitor (Takara Bio, Beijing, China). qPCR was performed in triplicate with RealStar Green Fast Mixture (A303, GenStar, Beijing, China) and on a StepOnePlus thermal cycler (ABI, Thermo Fisher, MA, USA). Threshold cycle numbers were normalized to triplicate samples amplified with primers specific to glyceraldehyde-3-phosphate dehydrogenase (*G**APDH*). qPCR primers for the target genes are listed in Table 1.

### 2.5. Immunoblotting

Immunoblotting assay was performed as previously described [27]. Cells were lysed on ice with a lysis buffer (50 mM Tris-Cl at pH 7.4, 150 mM NaCl, 1% Triton X-100, 1% sodium deoxycholate, 1 mM Na_3_VO_4_, 1 mM EDTA, and 1 mM PMSF) for 60 min. Whole-cell lysates were separated by sodium dodecyl sulfate–polyacrylamide gel electrophoresis (SDS–PAGE) and transferred to a polyvinylidene fluoride membrane (Millipore) for immunoblotting with specific antibodies.

### 2.6. Dual-Luciferase Reporter Assay

Human *RIG-I*, *MDA5*, or *LGP2* promoter sequences (from positions −1500 to 0) were amplified by DNA Polymerase (Vazyme Biotech Co., Ltd., Nanjing, China) and cloned into pGL3-basic vectors to generate RIG-I-luc, MDA5-luc, or LGP2-luc reporter plasmids. Putative p65-binding sites in *RIG-I*, *MDA5*, or *LGP2* promoters were predicted using the online software JASPAR (http://jaspar.genereg.net/ (accessed on 29 May 2022)). The mutants of *RIG-I* or *MDA5* promoters lacking putative p65-binding sites were amplified by overlap PCR using specific primers, and were cloned into pGL3-basic vectors. For luciferase assays, HEK293T cells were co-transfected with RIG-I-luc, MDA5-luc, and LGP2-luc, or RIG-mut-luc, MDA5-mut-luc reporter plasmids, and expressing plasmids encoding UL13 or empty vectors using Lipofectamine 2000 (Invitrogen, Carlsbad, CA, USA). Twenty-four hours after transfection, cell lysates were prepared and analyzed using the Dual-Luciferase Report Assay System (Promega, Madison, WI, USA), according to the manufacturer’s instructions. The Renilla luciferase reporter gene (pRL-TK, Promega) was used as an internal control.

### 2.7. Inhibition of Signaling Pathways

For blocking of NF-κB and MAPK signals, specific chemical inhibitors were applied as described previously [28]. Briefly, cells were pretreated with the NF-κB inhibitor Bay11-7082 (Bay11, 10 µM), JNK inhibitor SP600125 (SP, 10 µM), MEK/ERK inhibitor U0126 (10 µM), or p38 MAPK inhibitor SB203580 (SB, 10 µM) for 1 h prior to poly(I:C) (InvivoGen, San Diego, CA, USA) transfection. All chemical inhibitors were purchased from MedChemExpress (MCE, Monmouth Junction, NJ, USA).

### 2.8. Cytoplasmic and Nuclear Protein Extraction

Cells were lysed on ice with a lysis buffer (10 mM HEPES at pH 7.9, 50 mM NaCl, 0.5 mM Sucrose, 0.1 mM EDTA, 0.5% Triton X-100, 1 mM DTT, 10 mM Sodium pyrophosphate decahydrate, 0.5 M NaF, 0.2 M Na3VO4, 1 mM PMSF, and protease inhibitor mixture) for 60 min. The supernatant was collected for the cytoplasmic extract after centrifuging for 5 min at 1500 rpm. The pellet was resuspended with Buffer A (10 mM HEPES at pH 7.9, 10 mM KCl, 0.1 mM EGTA, 0.1 mM EDTA, 1 mM DTT, 1 mM PMSF, and protease inhibitor mixtures), centrifuged for 5 min at 1500 rpm, and then the supernatant was removed. Then, four volumes of buffer C (10 mM HEPES at pH 7.9, 500 mM NaCl, 0.1 mM EGTA, 0.1 mM EDTA, 0.1% Nonidet P-40, 1 mM DTT, 1 mM PMSF, and protease inhibitor mixtures) were added, and vortexed for 30 min at 4 °C. The supernatant was collected as the nuclear extract after centrifugation for 10 min at 14,000 rpm. β-actin and Lamin B1 were used as loading controls for the cytoplasm and nucleus, respectively.

### 2.9. Immunofluorescence

UL13-HEK293T cells or control cells were transfected with poly(I:C) for 3 h and then fixed for 20 min with 4% cold paraformaldehyde. Cells were then permeabilized for 10 min with 0.1% Triton X-100, and then blocked with 5% bovine serum albumin (BSA, Sigma) for 30 min. Cells were incubated with the appropriate primary antibodies for 120 min, followed by staining with Alexa Fluor 594-conjugated secondary antibodies (Proteintech Group Inc. China) or 647-conjugated secondary antibodies (TIANGEN, Beijing, China) for 60 min. Nuclei were stained with DAPI. Images were visualized and acquired using a laser scanning confocal microscope with LAS X software (Leica, Wetzlar, Germany).

### 2.10. Statistical Analysis

All statistical analyses were carried out using GraphPad Prism v8.0 software; *p*-values were calculated with a two-tailed paired or unpaired Student’s *t*-test, and *p*-values ≤ 0.05 were considered significant.

## 3. Results

### 3.1. UL13 Inhibits RLR-mediated Antiviral Immune Responses

The role of the PRV tegument protein UL13 has been indicated in the suppression of host antiviral immune responses [24,29]. To confirm the suppressive effect of UL13 on RLR-mediated IFN-I signaling, we generated stable HEK293T cells ectopically expressing Flag-tagged UL13 (UL13-HEK293T cells). Immunoblotting analysis showed that Flag-UL13 was successfully expressed in UL13-HEK293T cells (Figure 1A). Upon poly(I:C) transfection or Sendai virus (SeV) infection, which typically activate RLR-mediated immune responses, UL13-HEK293T cells showed impaired gene expression of *IFNB1* and downstream ISGs—including *ISG56*, *MX1*, and *OAS**L*—compared with control cells (Figure 1B,C), indicating that UL13 inhibits RLR-mediated expression of type-I IFN and downstream ISGs. This observation was further confirmed in PK-15 cells stably expressing UL13 (UL13-PK-15 cells) (Figure 1D,E). These results suggest that UL13 functions as an antagonist of RLR-mediated type-I IFN responses.

### 3.2. UL13 Inhibits RLR-Mediated Antiviral Response by Suppressing Transcription of RIG-I and MDA5

The transfected cytosolic RNA is mainly recognized by RLRs, and activates the adaptor protein MAVS to initiate innate antiviral immune response [11]. To know whether UL13 targets the RLR-mediated signaling pathway, we next examined the effect of UL13 on transfected poly(I:C)-mediated immune responses in HEK293T cells or PK-15 cells. The results showed that UL13 expression markedly inhibited the expression of RIG-I and MDA5, as well as phosphorylation of TBK1, in response to poly(I:C) transfection (Figure 2A), indicating that UL13 likely targets the expression of RIG-I-like receptors to suppress RNA-mediated immune response. Similarly, upon SeV infection, UL13 expression also inhibited expression of RLR receptors—including RIG-I and MDA5—and phosphorylation of TBK1 and IRF3 in both HEK293T cells and PK-15 cells (Figure 2B), further confirming the suppressive effect of UL13 on RNA-mediated immune responses. Notably, UL13 did not alter total protein levels of MAVS, TBK1, and IRF3 (Figure 2A,B), suggesting that UL13 possibly targets the upstream RLRs to suppress RLR-mediated signaling activation. Indeed, UL13 expression strikingly suppressed gene transcription of *RIG-I* and *MDA5* under poly(I:C) transfection conditions (Figure 2C). However, expression of UL13 did not affect mRNA expression of *LGP2*—another RIG-I like receptor. To confirm the effect of UL13 on RLR transcription, we first constructed luciferase reporter plasmids by introducing RIG-I, MDA5, or LGP2 promoter sequences into the pGL3-basic vector. Luciferase assays showed that UL13 expression remarkably restrained activation of *RIG-I* or *MDA5* promoters in HEK293T cells, but not that of the *LGP2* promoter (Figure 2D), which is consistent with the suppressive effect of UL13 on transcription of RIG-I and MDA5 (Figure 2C). Together, these data demonstrate that UL13 negatively regulates the RLR signaling pathway via suppression of the transcription of *RIG-I* and *MDA5*.

### 3.3. Enforced Expression of RIG-I and MDA5 Counteracts UL13-Suppressed Antiviral Immune Responses

Having known that UL13 expression inhibits transcription of RIG-I and MDA5, leading to repression of RNA-triggered induction of IFNs and downstream antiviral responses, we next examined whether UL13 directly targets the mRNA of RLRs. qPCR analysis showed that UL13 overexpression has no effect on exogenous RIG-I/MDA5 transcription (Figure 3A), suggesting that UL13 may repress endogenous transcription of RLRs. Then, we investigated whether overexpression of RIG-I or MDA5 could rescue the phenotype. As expected, enforced expression of RIG-I or MDA5 counteracted UL13-induced suppression of antiviral immune responses, including expression of *IFNB1* and downstream ISGs (*ISG56* and *OAS**L*), both at the basal level and in response to stimulation of transfected poly(I:C) (Figure 3B), showing that RIG-I and MDA5 are the key targets for UL13 during RLR-mediated immune responses. In line with this observation, overexpression of RIG-I or MDA5 also enhanced phosphorylation of TBK1 induced by transfected poly(I:C) in UL13-HEK293T cells, without altering the expression of total TBK1 (Figure 3C). Collectively, these data demonstrate that UL13 inhibits RLR-mediated antiviral immune responses via downregulation of RLR expression.

### 3.4. UL13 Overexpression Restrains NF-κB Activation to Modulate Transcription of RIG-I and MDA5

Next, we sought to investigate the mechanisms by which UL13 suppresses the transcription of RLRs and RLR-mediated antiviral gene expression. It has been established that activation of NF-κB and mitogen-activated protein kinases (MAPKs) modulates gene transcription. We therefore examined whether UL13 modulates the activation of NF-κB and/or MAPKs to regulate IFN responses induced by poly(I:C) transfection. qPCR analysis showed that inhibition of NF-κB by a chemical inhibitor compromised the expression of *IFNB1* and downstream ISGs such as *ISG56* and *MX1*, whereas inhibition of MAPKs did not alter the UL13-mediated suppression of *IFNB1* and ISGs induced by transfected Poly(I:C) in HEK293T cells (Figure 4A), indicating that UL13 regulates NF-κB activation to repress RLR-mediated IFN responses. Consistently, immunoblotting analysis also showed that UL13 overexpression significantly suppressed phosphorylation of p65 and retarded degradation of IκBα in HEK23T cells transfected with poly(I:C) (Figure 4B). Importantly, overexpression of UL13 strikingly decreased the protein expression of RIG-I under both basal and stimulated conditions (Figure 4B), implying that suppression of NF-κB by UL13 is related to RIG-I expression. Moreover, UL13 overexpression impaired poly(I:C)-induced nuclear translocation of p65 (Figure 4C,E) and mRNA expression of *IL6* and *TNF* (Figure 4F)—two key inflammatory cytokines that are dependent on NF-κB. Altogether, these results suggest that UL13 modulates NF-κB activation to suppress RLR expression and RLR-mediated antiviral immune responses.

To further clarify whether UL13 regulates transcription of *RIG-I* or *MDA5* by inhibiting activation of NF-κB, we next examined the effects of p65 on the activation of *RIG-I*, *MDA5*, and *LGP2* promoters. Luciferase assays showed that p65 significantly promoted the activity of *RIG-I* and *MDA5* promoters, but not the *LGP2* promoter (Figure 4G), demonstrating that NF-κB activation is critical for transcription of *RIG-I* and *MDA5*. Interestingly, *RIG-I* and *MDA5* promoters, but not the *LGP2* promoter, contain several putative p65-binding sites (Figure 4H). To confirm the direct role of NF-κB in the transcription regulation of *RIG-I* or *MDA5*, we constructed two mutant *RIG-I* and *MDA5* luciferase reporter plasmids in which two putative p65-binding sequences were deleted (Figure 4H). The results showed that while UL13 expression dramatically suppressed wild-type *RIG-I*-promoter-driven luciferase activity, UL13 failed to suppress mutant *RIG-I*-promoter-driven luciferase activity when the p65-binding motif between −421 and −412 was deleted (Figure 4I). Similarly, UL13 inhibited wild-type *MDA5* promoter activity but failed to suppress mutant *MDA5* promoter activation when the p65-binding sites were deleted (Figure 4I). Together, the above data suggest that the transcription of *RIG-I* and *MDA5* is dependent on NF-κB, and that UL13 suppresses transcription of *RIG-I* and *MDA5* by inhibiting NF-κB activation.

### 3.5. UL13 Deficiency Potentiates RLR-Mediated Antiviral Responses during PRV Infection

Next, we investigated the effects of UL13 on *RIG-**I* and *MDA5* expression during PRV infection in HEK293T cells. Compared with wild-type PRV (PRV-WT), HEK293T cells showed higher levels of *RIG-I* or *MDA5* when infected with the mutant PRV lacking UL13 (PRV-∆UL13) (Figure 5A). Consequently, PRV-∆UL13 infection enhanced the induction of *IFNB1* and downstream ISGs, as well as the key inflammatory mediators, including *IL6* and *TNF* (Figure 5B,C). Consistent with these results, the expression of RIG-I and MDA5, along with the phosphorylation levels of TBK1 and p65—but not the key adaptor protein MAVS—were higher in HEK293T cells infected with PRV-∆UL13 than in PRV-WT-infected cells (Figure 5D). Collectively, these data suggest that deficiency of UL13 promotes RLR-mediated antiviral immune responses during PRV infection.

## 4. Discussion

IFN-mediated innate immune responses are crucial in protecting the host from viral infections and activating adaptive immunity [30,31]. In response to viral infection, host cells activate multiple signaling cascades to disrupt viral replication when specific molecular components of viruses are recognized [32,33]. Generally, cytosolic DNA sensors, such as cGAS, play important roles in recognition of DNA viruses [34,35,36]. Interestingly, recent studies suggest that RLRs sense RNA intermediates and host-derived RNAs during DNA virus infection [37,38,39,40]. As a DNA virus, PRV can establish long-term infection by evading cGAS-STING- and TLR3-induced antiviral innate immune responses [7,24,29]. However, how PRV evades RLR-mediated immune responses remains obscure. Here, we show that the PRV tegument protein UL13 suppresses type-I IFN production and downstream ISGs induced by RNA stimulation, indicating that UL13 functions as a novel antagonist of RLR-mediated antiviral responses. Hence, our study provides evidence that UL13 plays important roles in PRV’s evasion of RLR-mediated antiviral innate immune responses.

In recognition of RNAs, RLRs activate the MAVS–TBK1–IRF3 axis to induce type-I IFN responses [21]. Viruses, such as alphaherpesviruses, have developed multitudinous strategies for the evasion of RLR-mediated signaling [21]. Among these strategies, targeting RIG-I and MDA5 receptors is an effective way to inhibit RLR-mediated signaling transduction. For example, the HSV-1 UL11 C-terminal RNA-binding domain binds to RIG-I and MDA5, resulting in inhibition of downstream signaling activation [22]. HSV-1 UL37 deamidates RIG-I, rendering it unable to sense viral RNA and, thus, blocking its ability to induce antiviral immune responses [23]. HSV-2 utilizes the virion host shutoff (vhs) protein to suppress *RIG-I* and *MDA5* expression [41]. Consistently, we found that PRV infection could also significantly inhibit the gene transcription of *RIG-I* and *MDA5*, but not *LGP2.* Mechanistically, the PRV tegument protein UL13 inhibits transcription of *RIG-I* or *MDA5* by suppressing activation of NF-κB. Notably, both RIG-I and MDA5 are ISGs, and their expression can be quickly induced upon viral infection and dsRNA stimulation [42,43]. As IFN-inducible genes, increases in RIG-I and MDA5 expression may amplify the effects of IFNs [44]. Therefore, targeting of RIG-I and MDA5 by PRV can be a novel and simple strategy to escape RLR-mediated antiviral innate immune response.

Gene expression is under tight control of transcription factors that bind to unique DNA enhancer/repressor elements [26]. For *RIG-I*, several transcription factors have been reported to be directly involved in its transcriptional regulation. For example, interferon regulatory factor-1 (IRF1) positively regulates interferon- or dsRNA-induced RIG-I transcription [45]. CCAAT/enhancer-binding protein beta (C/EBPβ) acts as a repressor element to bind the *RIG-I* promoter for RIG-I transcriptional inhibition [46]. In this study, we found a key p65-binding site in the *RIG-I* proximal promoter. Deletion of the p65-binding site resulted in dramatic loss of RIG-I promoter activation induced by p65, which is consistent with a previous report [46]. Unlike *RIG-I*, the molecular mechanism that induces *MDA5* expression has not been presented. Here, we identified that p65 expression noticeably induced *MDA5* promoter activity. Deletion of two key p65-binding sites abolished activity of the *MDA5* promoter induced by p65, indicating that p65 is also the key transcriptional factor for *MDA5* transcription. In contrast, no putative p65-binding site was found in the *LGP2* promoter, which is consistent with the finding that UL13 did not affect *LGP2* activation. UL13 is a serine/threonine protein kinase that targets several viral and cellular substrates [47]. Studies have shown that UL13 can participate in various processes depending on its kinase activity. For example, UL13 directly modulates the phosphorylation of viral proteins VP11/12, ICP22, and UL49 [48,49], promotes the assembly and release of mature infectious virions [50], and stabilizes the viral ICP0 protein against degradation. Furthermore, UL13 targets STING or IRF3 to escape IFN-mediated antiviral innate immune response [24,25,29]. Here, we found that UL13 inhibited the expression of RIG-I and MDA5 by blocking the activation of NF-κB. However, the detailed mechanisms of how UL13 prevents NF-κB activation require further investigation.

In summary, this study identifies novel evasion strategies for PRV via suppression of the transcription of *RIG-I* and *MDA5* by UL13 through targeting NF-κB activation, further highlighting the importance of UL13 in the regulation of host innate antiviral immune responses.

## Figures and Tables

**Figure 1 viruses-14-01465-f001:**
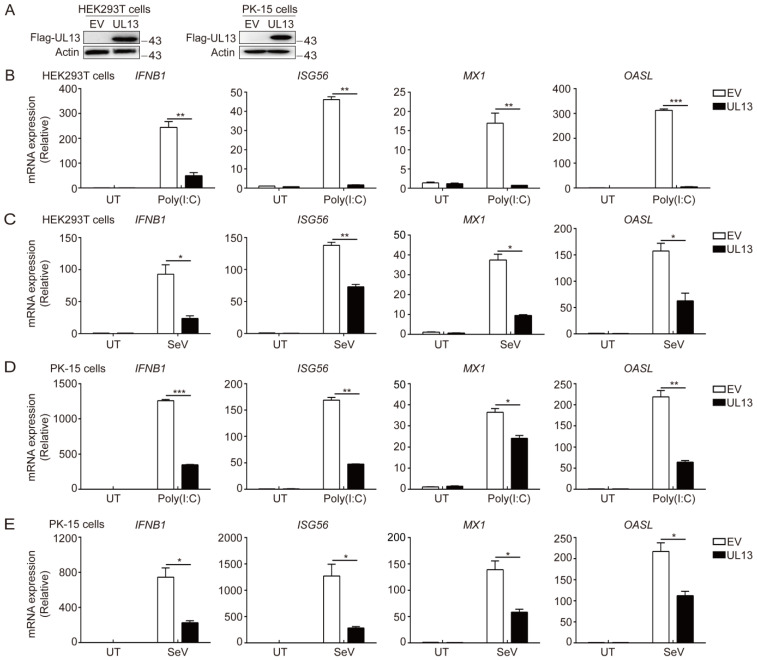
PRV UL13 negatively regulated RLR-mediated expression of type-I IFN and ISGs. (**A**) The expression of Flag-tagged UL13 in HEK293T or PK-15 cell lines was verified by immunoblotting; β-actin served as a loading control. (**B**–**E**) Quantitative real-time PCR (qPCR) analysis of *IFNB1* and downstream ISG (*ISG56*, *MX1*, and *OASL*) mRNA expression in UL13-HEK293T cells (**B**,**C**) or UL13-PK-15 cells (**D**,**E**), and control cells left untreated (UT) or transfected with 1 µg/mL of poly(I:C) (**B**,**D**) for 6 h or infected with SeV (MOI = 1) (**C**,**E**) for 12 h. Data are pooled from three independent experiments (mean ± SEM); *, *p* < 0.05; **, *p* < 0.01; ***, *p* < 0.001 (Student’s *t*-test).

**Figure 2 viruses-14-01465-f002:**
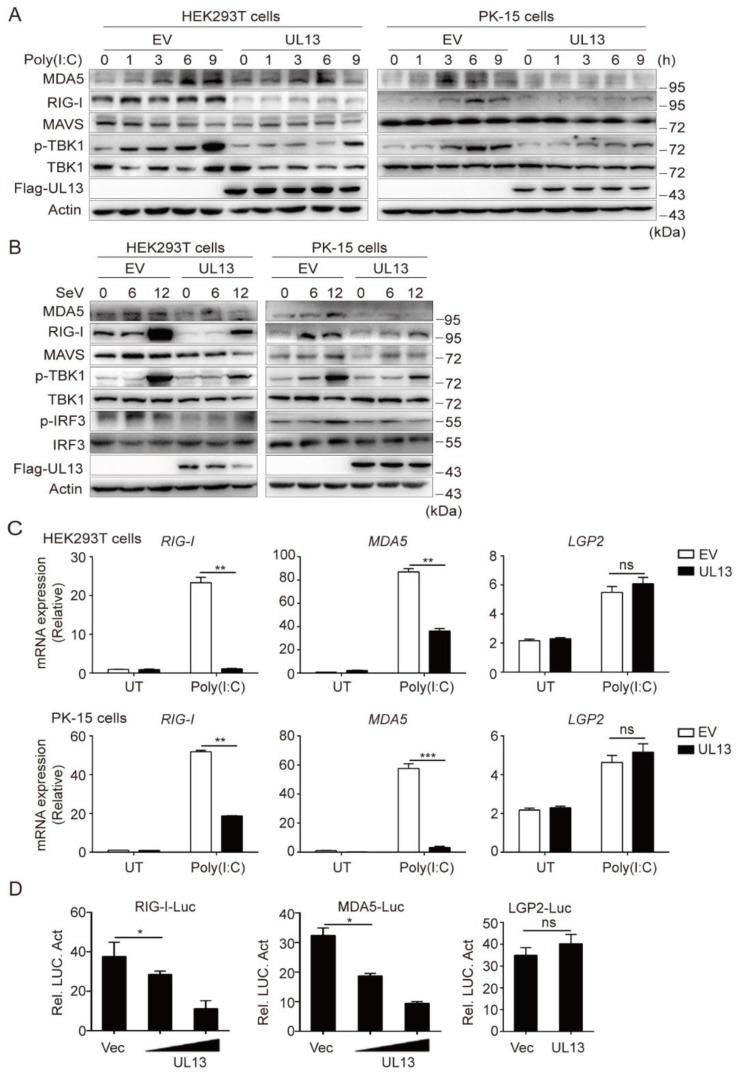
UL13 inhibits transcription of *RIG-I* and *MDA5* to suppress RLR-mediated antiviral responses. (**A**,**B**) Immunoblotting analysis of RIG-I, MDA5, MAVS, phosphorylated (Ser172) and total TBK1, and phosphorylated (Ser396) and total IRF3 in whole-cell lysates of UL13-HEK293T or UL13-PK-15 cells and control cells stimulated with poly(I:C) (**A**) or infected with SeV (MOI = 1, B) for the indicated times (above lanes); β-actin served as a loading control. (**C**) qPCR analysis of the mRNA levels of *RIG-I*, *MDA5*, and *LGP2* in UL13-HEK293T cells, UL13-PK-15 cells, or control cells left untreated (UT) or transfected with poly(I:C) for 6 h. (**D**) Luciferase activities in HEK293T cells co-transfected with *RIG-I*, *MDA5*, or *LGP2* promoter-driven luciferase reporters (50 ng) and plasmids encoding UL13 (concentration 150 ng, 300 ng) or empty vectors. Twenty-four hours after transfection, cell lysates were analyzed for luciferase activity. Data are representative of three independent experiments (**A**,**B**), or are pooled from three independent experiments (**C**,**D**, mean ± SD); *, *p* < 0.05; **, *p* < 0.01; ***, *p* < 0.001; ns, not significant (Student’s *t*-test).

**Figure 3 viruses-14-01465-f003:**
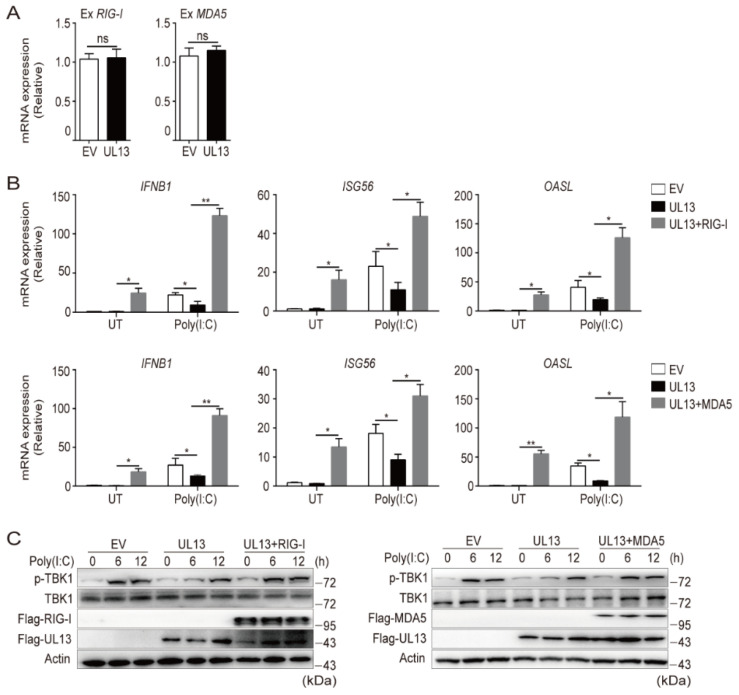
Enforced expression of RIG-I or MDA5 rescues UL13-suppressed antiviral immune responses. (**A**) qPCR analysis of *RIG-I* and *MDA5* mRNA expression in UL13-HEK293T cells or control cells transfected with plasmids encoding Flag-RIG-I or Flag-MDA5 for 24 h. (**B**) qPCR analysis of *IFNB1* and downstream ISG (*ISG56* and *OASL*) mRNA expression in cells transfected with plasmids encoding Flag-RIG-I, Flag-MDA5, or an empty vector for 24 h following poly(I:C) transfection for 6 h. (**C**) Immunoblotting analysis of phosphorylated (Ser172) and total TBK1, Flag-RIG-I or Flag-MDA5, and Flag-UL13 in UL13-HEK293T cells transfected with plasmids encoding Flag-RIG-I, Flag-MDA5, or an empty vector for 24 h following poly(I:C) transfection for the additional indicated periods. Data are pooled from three independent experiments (**A**,**B**, mean ± SD) or representative of three independent experiments (**C**); *, *p* < 0.05; **, *p* < 0.01; ns, not significant (Student’s *t*-test).

**Figure 4 viruses-14-01465-f004:**
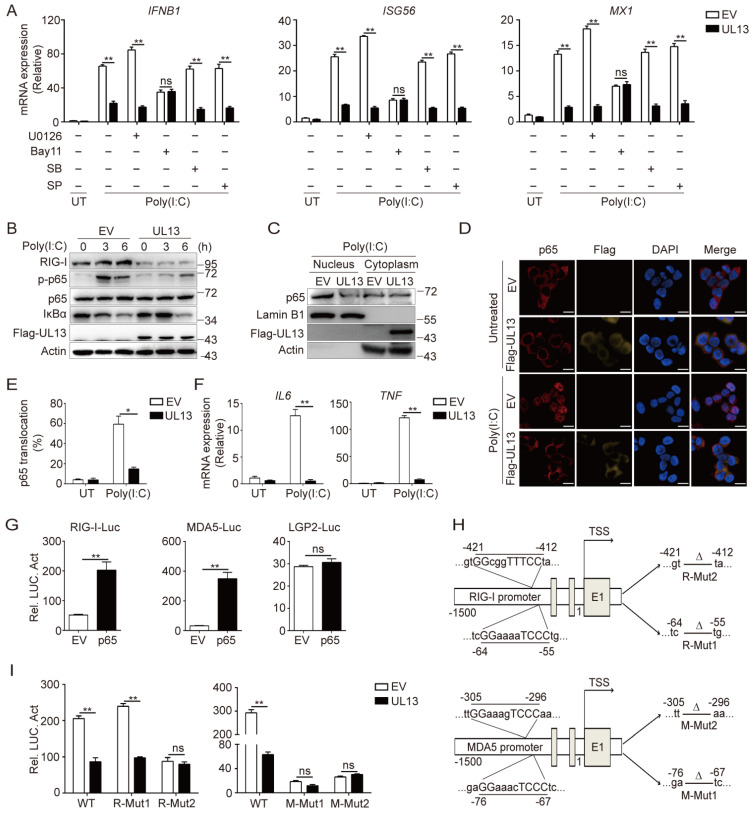
UL13 represses transcription of *RIG-I* and *MDA5* by regulating NF-κB activation. (**A**) qPCR analysis of mRNA expression of *IFNB1* and downstream ISG (*ISG56* and *MX1*) in UL13-HEK293T cells or control cells. Cells were left untreated or transfected with poly(I:C) for 6 h, or pretreated with NF-κB inhibitor (BAY11-7082, 10 µM), JNK inhibitor (SP600125, 10 µM), MEK/ERK inhibitor (U0126, 10 µM), or p38 inhibitor (SB203580, 10 µM) for 1 h, followed by poly(I:C) stimulation for 6 h. (**B**) Immunoblotting analysis of RIG-I, phosphorylated (Ser356) and total p65, IκBα, and Flag-UL13 in whole-cell lysates of UL13-HEK293T cells or control cells transfected with or without poly(I:C) for the indicated times. (**C**) Immunoblotting analysis of p65 protein levels in the nucleus and cytoplasm in UL13-HEK293T cells and control cells transfected with poly(I:C) for 3 h; β-actin and Lamin B1 served as the loading controls for the cytoplasm and nucleus, respectively. (**D**) Immunofluorescence analysis of the nuclear translocation of p65 in UL13-HEK293T cells or control cells left untreated or transfected with poly(I:C) for 3 h. Scale bars = 20 μm. (**E**) Quantification of nuclear localization of p65. (**F**) qPCR analysis of mRNA expression of *IL6* and *TNF* in cells as in (D). (**G**) Luciferase activities in UL13-HEK293T cells or control cells co-transfected with *RIG-I-*, *MDA5-*, or *LGP2*-promoter-driven luciferase reporters and plasmids encoding p65 or empty vectors. Twenty-four hours after transfection, cell lysates were analyzed for luciferase activity. (**H**) Annotation of putative p65-binding sites in *RIG-I* promoter or *MDA5* promoter and their mutants. (**I**) Luciferase activities in UL13-HEK293T cells or control cells co-transfected with *RIG-I-* or *MDA5*-promoter-driven luciferase reporters or mutants with plasmid encoding p65 or empty vectors. Twenty-four hours after transfection, cell lysates were analyzed for luciferase activity. Data are pooled from three independent experiments (**A**,**E**–**G**,**I**, mean ± SD) or representative of three independent experiments (**B**,**D**); *, *p* < 0.05; **, *p* < 0.01; ns, not significant (Student’s *t*-test).

**Figure 5 viruses-14-01465-f005:**
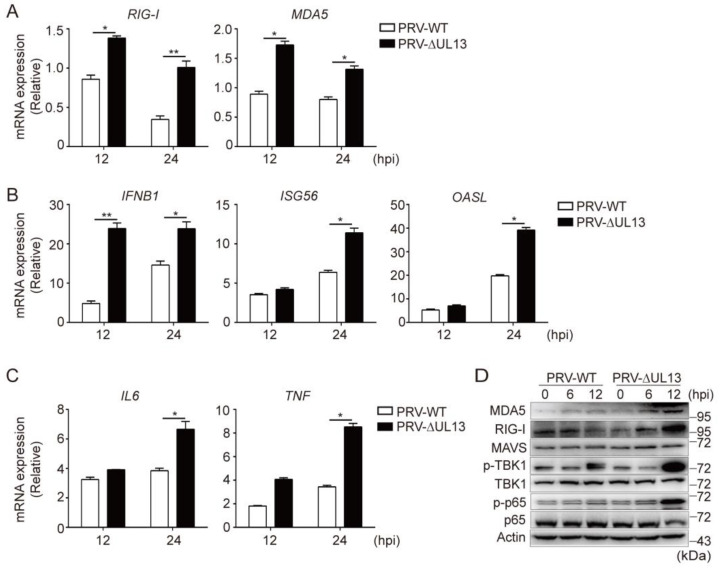
UL13 deficiency increases expression of RLRs and promotes RLR-mediated antiviral responses. (**A**) qPCR analysis of *RIG-I* and *MDA5* mRNA levels in HEK293T cells infected with PRV-WT or PRV-∆UL13 (MOI = 1) for the indicated times. (**B**,**C**) qPCR analysis the mRNA levels of *IFNB1*, downstream ISGs (*ISG56* and *OASL*) (**B**), and expression of *IL6* and *TNF* (**C**) in HEK293T cells infected with PRV-WT or PRV-∆UL13 (MOI = 1) for the indicated periods. (**D**) Immunoblotting analysis of MDA5, RIG-I, MAVS, phosphorylated and total TBK1, and phosphorylated and total p65 in HEK293T cells infected with PRV-WT or PRV-∆UL13 (MOI = 1). Data are pooled from three independent experiments (A, B, C, mean ± SD) or representative of three independent experiments (**D**); *, *p* < 0.05; **, *p* < 0.01; ns, not significant (Student’s *t*-test).

**Table 1 viruses-14-01465-t001:** Primers or sequences used in this study.

Gene	Forward Sequence (5′–3′)	Reverse Sequence (5′–3′)
	**Primer sequences for qPCR**	
*hGAPDH*	ATCAAGAAGGTGGTGAAGCA	GTCGCTGTTGAAGTCAGAGGA
*hIFNB1*	GCACTGGCTGGAATGAGACT	CCTTGGCCTTCAGGTAATG
*hISG56*	TTCGGAGAAAGGCATTAGA	TCCAGGGCTTCATTCATAT
*hMX1*	AGCCACTGGACTGACGACTT	ACCACGGCTAACGGATAAG
*hOASL*	CCCTGGGGCCTTCTCTTC	TCCTAACAGTGCCATTCCCT
*hTNF*	AATAGGCTGTTCCCATGTAGC	AGAGGCTCAGCAATGAGTGA
*hIL-6*	TAATGGGCATTCCTTCTTCT	TGTCCTAACGCTCATACTTTT
*h-RIG-I*	CTGGTTCCGTGGCTTTTTGG	CACCTGCCATCATCCCCTTA
*h-RIG-I-Flag*	GACTACAAGGACGACGATGA	AGTGTGGCAGCCTCCATTGG
*h-MDA5*	AAAGCTCCTACCCGAGTGTG	GCTGCCCACTTAGAGAAGCA
*h-MDA5-Flag*	GACTACAAGGACGACGATGA	AGGCTCCACCTGGATGTACA
*h-LGP2*	CAGCTGAGCCGACTTAGGAA	CGCAGCAGCAGTACTTAACC
*pGAPDH*	TACACTGAGGACCAGGTTGTG	TTGACGAAGTGGTCGTTGAG
*pIFNB1*	TGCATCCTCCAAATCGCTCT	ATTGAGGAGTCCCAGGCAAC
*pISG56*	TCCGACACGCAGTCAAGTTT	TGTAGCAAAGCCCTGTCTGG
*pMX1*	GCTTTCAGATGCTTCGCAGG	TGTCGTATGGCTGATTGCCT
*pOASL*	CAGGCCAACAGGTTCAGACAG	CAGGAAACCGCAGACGATGT
*pTNF*	CGACTCAGTGCCGAGATCAA	CTCACAGGGCAATGATCCCA
*pIL6*	AAGCTGCAGTCACAGAACGA	GGACGGCATCAATCTCAGGT
*p-RIG-I*	TCCTTCTGACTGCTAACGCT	ACTAAGGAAGGTGTCCAGCAG
*p-MDA5*	TGAGGACTGATGTTTGATTCCAG	ACCTCTGCCCACCAAGATAGA
*pLGP2*	CAGCCCTGCAAACAGTACGAC	CACTCCAGTTTCGGGTTCTC
	**Primer sequences for regular PCR**	
PRV UL13	CCGGAATTCATGGACTACAAAGACGATG	CGCGGATCCTCAGGCATCGAGTTCG
*hRIG-I*-Luc	CGGGGTACCAAGTTTATCTGTAGGTTCAATG	GGAAGATCTGCCTCACTAGCTTTAAAGCC
*hMDA5*-Luc	CGGGGTACCCCAAGGTTTCATTTACTTCAAC	GGAAGATCTCCTGACTTTGGTTTCTGTTT
*hLGP2*-Luc	CGGGGTACCGGAGACCAGGTTTCCTTTCCAG	GGAAGATCTAGAAATGGAAACTGAAACTGAG
*hRIG-I*-Luc-M1	ATTTGGACAACAGGTTATAAAGCTAAACAT	ATGTTTAGCTTTATAACCTGTTGTCCAAAT
*hRIG-I*-Luc-M2	TTTGGACAACAGGTTATAAAGCTAAACAT	ATGTTTAGCTTTATAACCTGTTGTCCAAA
*hMDA5*-Luc-M1	GCCTGGCGGGGATCAGGGAGACGC	GCGTCTCCCTGATCCCCGCCAGGC
*hMDA5*-Luc-M2	AGCATGTGATTTAAAGGGGAAGTG	CACTTCCCCTTTAAATCACATGCT

## Data Availability

All analyzed data are contained in the main text. Raw data are available from the authors upon request.

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
