# Peer review of "Pseudorabies Virus Tegument Protein UL13 Suppresses RLR-Mediated Antiviral Innate Immunity through Regulating Receptor Transcription"

_viruses, 2022, doi:10.3390/v14071465_

Round 1
Reviewer 1 Report
The work entitled 'Pseudorabies Virus UL13 Suppresses RLR-mediated Antiviral 3 Innate Immunity through Regulating Receptor Transcription' by Zhao et al., is overall an interesting study.
The major concern I have is that the authors have not elucidated the mechanism fully. They conclude that UL13 suppresses RLR-mediated signaling by blocking NF-kB activation. However, NF-kB is an extensive pathway with multiple players. More experiments in this regard are needed too exactly determine the important factors needed for UL-13 mediated suppression.
Does over expression of UL-13 in the cells impact its growth?
The figure quality should be improved, western blot in Figure 1 should be an independent panel.
Scale bars are missing in Figure 4D.
Author Response
We thank the reviewers for their time and insightful and constructive comments. We are pleased to know that the reviewer felt that “it is novel in that UL13 can operate to this effect”, and “it can be a valuable contribution to the field”. Reviewers also felt that “this is an interesting and novel study”. We have experimentally addressed the points raised by the reviewers and have substantially revised the manuscript by adding new figures panels as well as clarifying writing. The reviewers’ points are specifically addressed below. Changes in the manuscript have been marked in red.
Major Issues:
Point 1: The major concern I have is that the authors have not elucidated the mechanism fully. They conclude that UL13 suppresses RLR-mediated signaling by blocking NF-kB activation. However, NF-kB is an extensive pathway with multiple players. More experiments in this regard are needed too exactly determine the important factors needed for UL-13 mediated suppression.
Response 1: Thanks for the reviewer’s comments. In fact, we have performed experiments to identify the interaction proteins with UL13 in HEK293T cells by LC-MS/MS. Although we still don’t know the exact mechanism that how does UL13 target NF-kB activation, our preliminary data analysis indicate that UL13 likely interact the NF-κB repressing factor (NKRF), a well-established inhibitor of p65 activation (data not shown). Hence, it is possible that UL13 may suppress NF-kB activation through interaction with NKRF. Further study will be performed to investigate the exact mechanisms for UL13-mediated suppression.
Point 2: Does over expression of UL-13 in the cells impact its growth?
Response 2: We count the cell numbers of UL13-HEK293T cells and its controls cells at multiple periods and found that UL13 expression did not impair cell growth (data not shown).
Point 3: The figure quality should be improved, western blot in Figure 1 should be an independent panel.
Response 3: As suggested, the western blot in Figure 1 was set as an independent panel (new Figure 1A) and all high-quality figures were uploaded accordingly (new Figure 1 to Figure 5).
Point 4: Scale bars are missing in Figure 4D.
Response 4: The scale bars have now been added in Figure 4D (new Figure 4D).
Reviewer 2 Report
It has been previously described that PRV tegument protein UL13 (a serine/threonine protein kinase) acts as an antagonist of antiviral immune responses via mutiple strategies, such as promoting ubiquitination and degradation of IRF3 (Lv, et al. 2020) and STING (Kong, et al. 2022), phosphorylating IRF3 (Bo, et al. 2020). In this manuscript, the authors have revealed that UL13 also functions as an antagonist of RLR-mediated antiviral immune responses through inhibiting NF-κB activation to suppress transcription of RLR receptor (RIG-I and MDA5). It is novel in that UL13 can operate to this effect. Overall, it can be a valuable contribution to the field. However, several points require attention and should be addressed as described below.
Major Issues
1. In Figure 2A, it is clear that UL13 overexpression significantly inhibited RIG-I expression with or without poly(I:C) transfection. But how about MDA5? MDA5 WB result should be included in Figure 2A, regardless of the change in MDA5 expression. Aslo, MDA5 WB result should be included in Figure 2B and Figure 5D. Then please delete Figure 2C because it is redundant with respect to Figure 2B (left panel). How to explain attenuated UL13 expression in HEK293T cells but not PK-15?
2. Based on the topic that UL13 suppresses RLR receptor (RIG-I and MDA5) transcription through inhibiting NF-κB, exogenous RLR receptor transcription should not be affected by UL13. Please add RIG-I and MDA5 qPCR result in Figure 3A in order to better support the topic.
3. It shows that there is no UL13 protein in nucleus in Figure 4C. But UL13 protein exists in the nucleus from Figure 4D. This needs to be clarified. Futhermore, UL13-HEK293T cells should be used in Figure 4D but two cells in the bottom left picture have no UL13 protein. It is strange for a stable cell line. Similar result occurs in Figure 2B (flag-UL13). The manuscript says that Luciferase assays showed that p65 significantly promoted the activity of RIG-I and MDA5 promoters, but not LGP2 promoter (Figure 4F), demonstrating that NF-κB activation is critical for transcription of RIG-I and MDA5. But there is no any data for LGP2 promote in Figure 4F. Please add it.
Minor Issues
Please check the full text of errors, and will not raise them one by one in the future.
1. line 2 Tegument Protein should be included in the title.
2. line 13 PRV-encoded, not PRV encoded
3. line 14, 264 RIG-I-like receptor, not RIG-1-like receptor
4. line 16, 18, 20 overexpression, not expression
5. line 26 Keywords should be: Pseudorabies Virus (PRV); Tegument protein UL13; RIG-I; MDA5; NF-κB
6. line 36 pattern recognition receptors (PRRs)-induced
7. line 60-62 Among them, tegument proteins have a wide variety of functions in viral entry, secondary envelopment and viral capsid transportation during infection and immune escape.
8. line 144 what's the source of poly I:C? Please note.
9. line 205 remove (Figure 2A)
10. line 211-212 The manuscript says that "Indeed, UL13 expression strikingly suppressed gene transcription of 212 RIG-I and MDA5 under Poly(I:C) transfection or SeV infection conditions (Figure 2D)". But there is no data from SeV infection in Figure 2D.
11. line 232 please note plasmid amount used in Figure 2E.
12. line 245 OASL? It should be OAS1 based on Figure 3A and Table 1.
13. line 255 It should be OAS1.
14. line 274 NF-κB
15. line 308 ERK inhibitor should be replaced with MEK/ERK inhibitor. Also in line 143, MEK inhibitor should be replaced with MEK/ERK inhibitor. Please pay more attention to inhibitor concentration unit. It is different between in line 143 and 308.
16. line 317 in Figure 4F legend, please note scale bar.
17. line 358-359 viruses also modify cytosolic and nuclear signaling effectors for blocking this response
18. line 369-370 UL13-knockout PRV
Author Response
We thank the reviewers for their time and insightful and constructive comments. We are pleased to know that the reviewer felt that “it is novel in that UL13 can operate to this effect”, and “it can be a valuable contribution to the field”. Reviewers also felt that “this is an interesting and novel study”. We have experimentally addressed the points raised by the reviewers and have substantially revised the manuscript by adding new figures panels as well as clarifying writing. The reviewers’ points are specifically addressed below. Changes in the manuscript have been marked in red.
Major Issues
Point 1: In Figure 2A, it is clear that UL13 overexpression significantly inhibited RIG-I expression with or without poly(I:C) transfection. But how about MDA5? MDA5 WB result should be included in Figure 2A, regardless of the change in MDA5 expression. Aslo, MDA5 WB result should be included in Figure 2B and Figure 5D. Then please delete Figure 2C because it is redundant with respect to Figure 2B (left panel). How to explain attenuated UL13 expression in HEK293T cells but not PK-15?
Response 1: According to the reviewer’s suggestion. We performed additional experiments to examine whether UL13 overexpression affects MDA5 expression. The results showed that UL13 overexpression also inhibited the expression of MDA5 in both HEK239T cells or PK-15 cells upon Poly(I:C) transfection or SeV infection (new Figure 2A and B, described on page 6, line 208). We also examined the MDA5 expression in HEK293T cells infected with by PRV-WT or PRV-∆UL13. The results showed that the protein expression of MDA5 in HEK293T infected with PRV-∆UL13 was higher than that of cells infected with PRV-WT, further supporting that deficiency of UL13 benefits to MDA5 expression (new Figure 5D, described on page 13, line 349). As suggested, the original Figure 2C has been removed.
Point 2: Based on the topic that UL13 suppresses RLR receptor (RIG-I and MDA5) transcription through inhibiting NF-κB, exogenous RLR receptor transcription should not be affected by UL13. Please add RIG-I and MDA5 qPCR result in Figure 3A in order to better support the topic.
Response 2: We thank the reviewer for raising this constructive suggestion. We therefore performed experiments to examine the effect of UL13 on exogenous RLR receptor transcription in HEK293T cells by transfection of Flag-tagged RIG-I and MDA5 expression plasmids. Indeed, qPCR analysis that UL13 expression did not affect the transcription of Flag-tagged RIG-I and MDA5 transcription, supporting that UL13 suppresses RLR receptor transcription through inhibiting NF-κB activation (new Figure 3A, described on page 9, line 248-251).
Point 3: It shows that there is no UL13 protein in nucleus in Figure 4C. But UL13 protein exists in the nucleus from Figure 4D. This needs to be clarified. Futhermore, UL13-HEK293T cells should be used in Figure 4D but two cells in the bottom left picture have no UL13 protein. It is strange for a stable cell line. Similar result occurs in Figure 2B (flag-UL13). The manuscript says that Luciferase assays showed that p65 significantly promoted the activity of RIG-I and MDA5 promoters, but not LGP2 promoter (Figure 4F), demonstrating that NF-κB activation is critical for transcription of RIG-I and MDA5. But there is no any data for LGP2 promote in Figure 4F. Please add it.
Response 3: Thanks for the reviewer to raise these valid concerns. In Figure 4C, the experiments were done in UL13-HEK293T stable cells (Figure 4C). In contrast, in the original Figure 4D, UL13 was expressed in HEK293T by transient transfection, which was described incorrectly in the original Figure legends. Indeed, overexpression UL13 was mostly localized in nucleus in HEK293T cells (Bo et al., 2021). As suggested, we re-performed the immunofluorescence experiments in UL13-HEK293T stable cells. The results showed that expression of UL13 was mainly localized in cytoplasm (new Figure 4D), which is consistent to the results in Figure 4C.
According to the reviewer’s suggestion, we performed additional luciferase assays to examine the effect of p65 on LGP2 promoter and found that p65 expression did not alter the activity of LGP2 promoter, indicating that NF-κB activation is likely not critical for transcription of LGP2 (new Figure 4G, right panel).
Minor Issues
Point : Please check the full text of errors, and will not raise them one by one in the future.
Response: We appreciate all the comments from the reviewer. As suggested, we carefully checked the entire text of errors and corrected it wherever possible. All the indicated errors have been corrected accordingly.
Point 1: line 2 Tegument Protein should be included in the title.
Response 1: “Tegument Protein” is now included in the title (page 1, line 2).
Point 2: line 13 PRV-encoded, not PRV encoded.
Response 2: This error has been corrected in the text (page 1, line 14).
Point 3: line 14, 264 RIG-I-like receptor, not RIG-1-like receptor.
Response 3: “RIG-I-like receptor” are now used in the whole manuscript (page 1, line 15; page 10, line 276).
Point 4: line 16, 18, 20 overexpression, not expression.
Response 4: The word “overexpression” was used in the text (page 1, line 17, 19, 21).
Point 5: line 26 Keywords should be: Pseudorabies Virus (PRV); Tegument protein UL13; RIG-I; MDA5; NF-κB.
Response 5: The above key words were used in the revised manuscript (page 1, line 27).
Point 6: line 36 pattern recognition receptors (PRRs)-induced.
Response 6: This error has been corrected (page 1, line 37).
Point 7: line 60-62 Among them, tegument proteins have a wide variety of functions in viral entry, secondary envelopment and viral capsid transportation during infection and immune escape.
Response 7: As suggested, this sentence has been revised accordingly (page 2, line 61-63).
Point 8: line 144 what's the source of poly I:C? Please note.
Response 8: Poly(I:C) was purchased from InvivoGen. The vendor information has now been included (page 4-5, line 147-148).
Point 9: line 205 remove (Figure 2A).
Response 9: Removed.
Point 10: line 211-212 The manuscript says that "Indeed, UL13 expression strikingly suppressed gene transcription of RIG-I and MDA5 under Poly(I:C) transfection or SeV infection conditions (Figure 2D)". But there is no data from SeV infection in Figure 2D.
Response 10: Thanks for the reviewer to point out this issue. The sentence has been revised as following “Indeed, UL13 expression strikingly suppressed gene transcription of RIG-I and MDA5 under Poly(I:C) transfection (Figure 2C)” (page 7, line 216-217).
Point 11: line 232 please note plasmid amount used in Figure 2E.
Response 11: For RIG-I, MDA5 or LGP2 promoter-driven luciferase reporter constructs, 50 ng total amounts of plasmid DNA were transfected. For UL13 expression plasmids, 150 ng and 300 ng total amounts of plasmid DNA were used, respectively This information were now included in Figure legends (page 9, line 238-239).
Point 12: line 245 OASL? It should be OAS1 based on Figure 3A and Table 1.
Response 12: The name of this gene is OASL or OASL1. We have corrected the errors in the text (page 5, line 185; page 9, line 254; page 10, line 267; page 14, line 359).
Point 13: line 255 It should be OAS1.
Response 13: Please see the above response.
Point 14: line 274 NF-κB.
Response 14: The typing error has been corrected (page 11, line 282).
Point 15: line 308 ERK inhibitor should be replaced with MEK/ERK inhibitor. Also in line 143, MEK inhibitor should be replaced with MEK/ERK inhibitor. Please pay more attention to inhibitor concentration unit. It is different between in line 143 and 308.
Response 15: We thank the reviewer’s comments. We replaced “ERK inhibitor” with “MEK/ERK inhibitor” (page 12, line 321; page 4, line 146). All chemical inhibitors were used at a final concentration of 10 μM. It is corrected in the manuscript (page 12, line 320-322).
Point 16: line 317in Figure 4F legend, please note scale bar.
Response 16: Scale bar information has been noted in the Figure 4F legend (page 13, line 330).
Point 17: line 358-359 viruses also modify cytosolic and nuclear signaling effectors for blocking this response.
Response 17: The sentence has been revised according to the reviewer’s suggestion (page 14, line 373-374).
Point 18: line 369-370 UL13-knockout PRV.
Response 18: Revised (page 14, line 384).
Reviewer 3 Report
Zhao et al submitted to viruses an interesting study focusing on the function of UL13 and how it regulates the transcription of RIG-I and MDA5 and suppresses RLR-mediated signals for immune evasion. Using stable HEK293T/PK-15 cells expressing UL13, the authors found that ectopically expressed UL13 diminishes Poly(I:C) transfection-or Sendai virus (SeV) infection-induced production of type I IFN and downstream ISGs. Importantly, they proved that UL13 suppresses the transcription levels of RIG-I and MDA5 by targeting the promoter activities of those receptors using powerful luciferase assays. In mechanism, UL13 affects p65 translocation and modulates NF-κB activities to regulate RLR expression and impair RLR-mediated inflammatory responses. Overall, this is an interesting and novel study, which demonstrates the essential role of UL13 in the downregulation of RLR receptors’ expression and for escaping PRV from antiviral innate immune responses.
Major
1. In figure 4B, did the authors also check the expression level of MDA5?
2. In figure 4D, the confocal image of translocation of p65 is not clear. A better figure is needed to support the conclusion. It might be good to account for more cells and do a statistical analysis of the translocated p65.
3. In figure 5, did the depletion of UL13 affect PRV replication?
Minor Issues or suggestions
1. In line 246, “Figure 3B” should be “Figure 3A”
2. In line 250, “Figure 3C” should be “Figure 3B”
3. In lines 52-54 and 57. The authors made a statement “Studies found that DNA viruses replication generates RNA intermediates, which could be nucleic acid ligands for RLRs”. For recognition of RNA intermediates from DNA viruses by RLR and Virus evolved numerous strategies to evade them, a recently published paper” Battle Royale: Innate Recognition of Poxviruses and Viral Immune Evasion (PMID: 34356829)” can be cited. This citation will help support the authors’ statement.
Author Response
We thank the reviewers for their time and insightful and constructive comments. We are pleased to know that the reviewer felt that “it is novel in that UL13 can operate to this effect”, and “it can be a valuable contribution to the field”. Reviewers also felt that “this is an interesting and novel study”. We have experimentally addressed the points raised by the reviewers and have substantially revised the manuscript by adding new figures panels as well as clarifying writing. The reviewers’ points are specifically addressed below. Changes in the manuscript have been marked in red.
Major
Point 1: In figure 4B, did the authors also check the expression level of MDA5?
Response 1: We thank the reviewer’s comments. Although we did not check the expression level of MDA5 in Figure 4B, we performed additional immunoblotting analysis to examine the MDA5 expression levels in Figure 2A and 2B. The results showed that UL13 overexpression also inhibited MDA5 protein levels in both HEK293T cells and PK-15 cells in response to Poly(I:C) transfection or SeV infection (new Figure 2 A and B, described on page 6), confirming the suppressive role of UL13 on both MDA5 and RIG-I.
Point 2: In figure 4D, the confocal image of translocation of p65 is not clear. A better figure is needed to support the conclusion. It might be good to account for more cells and do a statistical analysis of the translocated p65.
Response 2: To better support the conclusion, we performed new experiments and provided better images (new Figure 4D). We also do statistical analysis(new Figure 4E)of the translocated p65 by counting.
Point 3: In figure 5, did the depletion of UL13 affect PRV replication?
Response 3: In our recent study, we did examine the effects of deficiency of UL13 on viral replication of both wild-type (PRV-WT) or UL13-deficient PRV (PRV-ΔUL13) and the mutant PRV(PRV-∆UL13) in multiple cells including mouse embryonic fibroblasts (MEFs). We observed similar propagation between wild-type or UL13-deficient PRV, indicating that UL13 is nonessential for PRV replication (Kong et al, PLoS Pathogens, 2022. Ref 24).
Minor issues or suggestions
Point 1: In line 246, “Figure 3B” should be “Figure 3A”.
Response 1: The error has now been corrected in the text (page 9, line 255).
Point 2: In line 250, “Figure 3C” should be “Figure 3B”.
Response 2: This error has been corrected in the text (page 9, line 259).
Point 3: In lines 52-54 and 57. The authors made a statement “Studies found that DNA viruses replication generates RNA intermediates, which could be nucleic acid ligands for RLRs”. For recognition of RNA intermediates from DNA viruses by RLR and Virus evolved numerous strategies to evade them, a recently published paper” Battle Royale: Innate Recognition of Poxviruses and Viral Immune Evasion (PMID: 34356829)” can be cited. This citation will help support the authors’ statement.
Response 3: The reviewer’s suggestion is appreciated. The paper mentioned above has been cited in the text now (Ref 18; page 2, line 55; page 16, line 500).
Round 2
Reviewer 1 Report
The authors have addressed my concern and the paper can be accepted.
Author Response
Response:We would greatly appreciate opinions from the reviewers on the revised manuscript. We have revised the manuscript carefully according to reviewer’s comments and extensively modified the languages in the text especially in Introduction and Discussion parts. Changes in the manuscript have been marked in red.
Reviewer 2 Report
It is well modified, but still has several mistakes.
1. line 73-74, RLR-mediated antiviral immune should be replaced with RLR-mediated antiviral innate immunity.
2. Please confirm Flag-MDA-5 protein size in Figure. 3C right panel.
Author Response
Point: It is well modified, but still has several mistakes.
Response: We would greatly appreciate opinions from the reviewers on the revised manuscript. As suggested, we carefully checked the entire text of errors and corrected it wherever possible. We also modified the languages in the text especially in Introduction and Discussion parts. Changes in the manuscript have been marked in red.
Point 1: line 73-74, RLR-mediated antiviral immune should be replaced with RLR-mediated antiviral innate immunity.
.Response 1: The sentence has been revised according to the reviewer’s suggestion (page 2, line 75).
Point 2: Please confirm Flag-MDA-5 protein size in Figure. 3C right panel.
Response:This error has been corrected in the Figure 3C.